# Therapeutic Targeting of Hypoxia-Inducible Factors in Cancer

**DOI:** 10.3390/ijms25042060

**Published:** 2024-02-08

**Authors:** Saba Musleh Ud Din, Spencer G. Streit, Bao Tran Huynh, Caroline Hana, Anna-Ninny Abraham, Atif Hussein

**Affiliations:** 1Department of Internal Medicine, Memorial Healthcare System, 703 North Flamingo Road, Pembroke Pines, FL 33028, USA; 2Department of Hematology and Oncology, Memorial Healthcare System, 703 North Flamingo Road, Pembroke Pines, FL 33028, USA; sstreit@mhs.net (S.G.S.); chana@mhs.net (C.H.); anabraham@mhs.net (A.-N.A.); ahussein@mhs.net (A.H.); 3Department of Pharmacy, Memorial Healthcare System, 703 North Flamingo Road, Pembroke Pines, FL 33028, USA

**Keywords:** cancer, hypoxia-inducible factors (HIFs), tumor microenvironment, transcription factors, epithelial–mesenchymal transition (EMT), signal transduction pathways, von Hippel–Lindau syndrome (VHL), vascular endothelial growth factor (VEGF), tumor immune escape

## Abstract

In the realm of cancer therapeutics, targeting the hypoxia-inducible factor (HIF) pathway has emerged as a promising strategy. This study delves into the intricate web of HIF-associated mechanisms, exploring avenues for future anticancer therapies. Framing the investigation within the broader context of cancer progression and hypoxia response, this article aims to decipher the pivotal role played by HIF in regulating genes influencing angiogenesis, cell proliferation, and glucose metabolism. Employing diverse approaches such as HIF inhibitors, anti-angiogenic therapies, and hypoxia-activated prodrugs, the research methodologically intervenes at different nodes of the HIF pathway. Findings showcase the efficacy of agents like EZN-2968, Minnelide, and Acriflavine in modulating HIF-1α protein synthesis and destabilizing HIF-1, providing preliminary proof of HIF-1α mRNA modulation and antitumor activity. However, challenges, including toxicity, necessitate continued exploration and development, as exemplified by ongoing clinical trials. This article concludes by emphasizing the potential of targeted HIF therapies in disrupting cancer-related signaling pathways.

## 1. Introduction

Cancer continues to pose an insurmountable challenge to healthcare systems worldwide, standing as one of the leading causes of mortality. Understanding the biological intricacies of cancer is essential for developing effective treatments and preventive strategies. One of the pivotal yet highly complex systems that has emerged as a significant focus in cancer research is the family of hypoxia-inducible factors (HIFs). These transcription factors serve as central regulators in maintaining cellular oxygen homeostasis and adaptively responding to low-oxygen environments, also known as hypoxia [1].

In the vast and intricate landscape of the human body, maintaining adequate oxygen levels for approximately 50 trillion cells is a colossal task. This becomes even more challenging considering the dramatically varying oxygen levels across different tissues and organs. HIFs play a crucial role in this delicate balancing act. They modulate the cellular metabolism and facilitate systemic responses like erythropoiesis and angiogenesis to ensure an adequate supply of oxygen when the need arises. This adaptive mechanism, although crucial for survival under physiological conditions, is usurped by cancer cells to gain a survival advantage in hypoxic environments commonly found in tumors [2].

Tumors have developed a notorious ability to adapt to and proliferate in hypoxic conditions, which would be lethal for normal cells. HIFs are crucial in enabling this adaptability. When activated in cancer cells, HIFs induce the transcription of hundreds to thousands of genes that not only shift the metabolic profile from oxidative phosphorylation to a more glycolytic phenotype but also promote other survival mechanisms, such as angiogenesis, thereby increasing the tumor’s blood supply. This makes HIFs a double-edged sword—vital for normal cellular function but potentially dangerous when co-opted by cancer cells [3].

The role of HIFs extends beyond the confines of cancer. For instance, their involvement in conditions like familial erythrocytosis and ischemic cardiovascular diseases reveals how perturbations in the HIF pathway can have wide-ranging implications. Such cases exemplify the intricate balance that HIFs maintain and how their dysregulation can result in pathophysiological conditions. These connections are not merely academic but hold substantial therapeutic implications [4].

As our understanding of the HIF pathway in cancer expands, so do the possibilities for clinical intervention. Researchers are increasingly looking at the HIF system as a potential target for therapeutic agents. These range from small molecules aimed at inhibiting HIF activity to gene therapies designed to modulate the expression of HIF-responsive genes. Furthermore, understanding the HIF pathway offers insights into the development of new diagnostic markers for early detection and prognosis of cancer [5].

In this review, we aim to unravel the convoluted relationships between HIFs and cancer. We will briefly discuss the regulation of HIF and its signaling pathway, dissect the molecular mechanisms that govern HIF activation and function, delve into their role in the tumor microenvironment, and explore the evolving landscape of HIF-targeted cancer therapies. This multi-faceted approach will offer readers a nuanced and thorough understanding of why and how HIFs are so integral in the pathology of cancer and what this means for future therapeutic strategies.

## 2. HIF Structure and Physiology

HIFs are essential transcription factors that respond to low oxygen conditions. They belong to the PER-ARNT-SIM (PAS) subfamily of the basic helix–loop–helix (bHLH) family. This group includes three major oxygen-labile HIF-α subunits (HIF-1α, HIF-2α, and HIF-3α) and a constitutive HIF-1β subunit (also known as aryl hydrocarbon receptor nuclear translocator, ARNT) [6].

HIF-1α, with 826 amino acids, is part of the HIF-1 complex. It belongs to the bHLH-PAS family, containing domains like PAS-A, PAS-B, HLH, and transactivation domains (N-TAD and C-TAD). The oxygen-dependent degradation (ODD) domain makes HIF-1α sensitive to oxygen levels. Under normoxia, hydroxylation of the ODD domain leads to ubiquitin-mediated degradation, while hypoxia inhibits hydroxylation, allowing HIF-1α to accumulate and initiate transcription [7].

While HIF-1 and HIF-2 share structural similarities and functions, they have unique target genes and distinct roles. HIF-1α is ubiquitously expressed, while HIF-2α is prominent in specific cell types [8]. For HIF-3α, there are multiple splice variants, with some lacking certain functional domains, leading to a variety of functions and interactions depending on the isoform. This complexity makes HIF-3α unique among the HIF-α subunits [9].

HIF-1 and HIF-2 are well characterized and have overlapping functions that promote cell survival, whereas HIF-3′s role remains less clear. The exact role and target genes of HIF-3α are still under investigation, but they may function differently from HIF-1α and HIF-2α, possibly even acting to negatively regulate some hypoxic responses [9].

The structures of HIF-1α, HIF-2α, HIF-3α, and HIF-1β can be seen in Figure 1.

## 3. HIF Regulation

HIFs regulate oxygen homeostasis transcriptionally in response to tissue hypoxia. The main subunits, HIF-1α, HIF-1β, HIF-2α, and HIF-3α, form heterodimeric proteins. Under normal oxygen conditions, hydroxylation by prolyl hydroxylases (PHD) and interaction with VHL protein lead to ubiquitination and degradation of HIF-α [4,10]. Refer to Figure 2.

Factor-inhibiting HIF (FIH) also regulates HIF-α via oxygen-dependent hydroxylation, inhibiting the interaction of HIF-α and CBP/p300 [11].

In hypoxic environments, there is reduced PHD activity, and hydroxylation is inhibited, leading to decreased degradation of HIF-α and increased accumulation, dimerization of corresponding HIF-β subunits, and translocation to cell nucleus [10]. Hydroxylation of the HIF-α subunit by FIH is also inhibited, which allows an increased expression of hypoxia-dependent genes by way of the HIF-α and CBP/p300 interaction [12].

There are many other factors, including reactive oxygen species, mitogen-activated protein kinase, nuclear factor κB (NF-κB), and more, which help to stabilize HIFs and increase translocation to the cell nucleus [10].

Lower levels of α-KG can inhibit the degradation of HIF-1α and enhance angiogenesis and tumorigenesis. This evidence suggests that the IDH-mutant-related decrease in α-KG stabilizes HIF-1α and leads to aberrant cellular proliferation [13].

## 4. Role of HIFs in Cancer Progression

Many types of cancers, including breast, cervical, head and neck cancers, and sarcoma, have been shown to contain regions of tumor hypoxia. In response to the hypoxia, there is an increased expression of HIF-1α and HIF-2α proteins [4]. These transcription factors orchestrate a complex network of genes that drive multiple facets of cancer progression, from angiogenesis and metabolic reprogramming to extracellular matrix (ECM) remodeling, epithelial–mesenchymal transition (EMT), cancer stem cell (CSC) specification, and immune evasion [4].

### 4.1. Regulation of Signal Transduction Pathways

Angiogenesis, the formation of new blood vessels from existing blood vessels [15], is a hallmark of cancer. HIFs are essential and participate in every step of angiogenesis. HIFs stimulate angiogenesis by upregulating genes that influence vascular endothelial growth factor (VEGF) expression and other pro-angiogenic factors, such as placenta-like growth factor, platelet-derived growth factor-β (PDGF-β), plasminogen activator inhibitor-1 (PAI-1), matrix metalloproteinases (MMP-2 and MMP-9), and angiopoietins (ANG-1 and ANG-2) [16]. The most potent pro-angiogenic factor is VEGF [17].

For example, HIF-1 binds to HREs on the VEGF promoter and stimulates increased gene transcription. HIF-1 also assists in stabilizing VEGF mRNA, and VEGF binds to different receptors (VEGFR) [18], resulting in new blood vessel formation, assisting tumor growth, and supplying tumors with essential nutrients and oxygen. The VEGF signaling cascade induced by HIF not only plays a vital role in angiogenesis but also plays a role in vascular endothelial cell (EC) biology. It was demonstrated by Skuli et al. that when HIF-2α function was ablated in murine vascular EC, there were defects in vessel structure and tumor angiogenesis [19].

Furthermore, HIF-1α has a role in regulating MMPs, PAI, and inducible nitric oxide synthase (iNOS), which have all been shown to promote angiogenesis [20]. MMPs are upregulated by HIF-1α-dependent pathways and are associated with extracellular matrix breakdown, allowing for the migration and proliferation of endothelial cells required for angiogenesis [21]. This has been shown to promote cancer invasion and growth [22]. PAI-1 expression is also upregulated by HIF-1α and assists in ECM protein breakdown, further contributing to the promotion of new blood vessels [23]. Additionally, HIF-1α also increases the expression of iNOS, which leads to an increase of nitric oxide (NO), which has been shown to increase angiogenesis and progression into metastasis in many different types of tumors [24].

HIFs favor anaerobic glycolysis over oxidative phosphorylation (OXPHOS) in cancer cells, a phenomenon known as the Warburg effect [25]. Glycolysis is an anaerobic pathway in which glucose is converted into pyruvate by a series of enzymatic reactions to yield ATP [25]. HIF-1 has been able to promote cell survival in hypoxic conditions by reprogramming glucose metabolism. HIF-1α increases the expression of glucose transporters, Glut1 and Glut3, and enzymes of the glycolytic pathway, including lactate dehydrogenase A (LDHA), phosphoglycerate kinase 1 (PGK-1) and hexokinases (HK1 and HK2) [26]. HIF-1 also works to suppress OXPHOS; both mechanisms lead to a shift in anaerobic glycolysis, leading to increased ATP production without the need for oxidative metabolism [27].

HIFs are crucial in EMT induction, allowing cancer cells to become more motile and invasive. EMT is a process of the transition of malignant cells from epithelial features to mesenchymal characteristics, leading to increased invasion, migration, and intravasation [28]. During this process, there is decreased epithelial cell-to-cell contact and polarity via the downregulation of E-cadherin. HIF-1 leads to the downregulation of E-cadherin by inducing transcription of factors such as Snail, TWIST, ZEB1, and TCF3 [29]. There is also an upregulation of mesenchymal markers such as Vimentin, N-cadherin, and α-smooth muscle actin, allowing for increased motility [30]. Intravasation is the invasion of cancer cells via the basement membrane into a blood or lymphatic vessel or circulating tumor cells (CTCs). CTCs are facilitated by HIF activation of the urokinase plasminogen activator surface receptor, hepatocyte growth factor receptor, and the MMPs [29]. Extravasation occurs when the CTC invades distant organs. This is thought to occur via chemokine activation through the HIF pathway. An example is the HIF-1-mediated L1 cell adhesion molecules and angiopoietin-like 4, which have been shown to be a stimulant in the extravasation of breast cancer to lung cancer [31].

They also play a vital role in specifying CSCs, which are characterized by self-renewal and tumor-initiating capabilities. These CSCs are primarily regulated by pluripotency factors like KLF4, OCT4, SOX2, and NANOG, and they present a challenge due to their resistance to conventional treatments.

### 4.2. Tumor Immune Escape and Immunotherapy Resistance

In the complex realm of cancer biology, tumor immune escape poses a significant obstacle to effective treatments. Hypoxic conditions, driven by inadequate blood supply as tumors grow, activate key players like HIF-1α and HIF-2α. These factors orchestrate immune escape by upregulating molecules like PD-L1, creating an immunosuppressive shield against CTLs [32]. HIFs also release molecules like VEGF, further suppressing the antitumor immune response. HIF-2α, less explored but linked to poor prognosis, merits investigation in immunotherapy resistance. HIFs’ impact extends beyond PD-L1, involving immune checkpoint regulators, MIC molecule shedding, upregulated cytokines like IL-23, and HIF-induced lncRNAs [32]. Unraveling these intricacies is vital for understanding tumor immune escape and developing targeted therapies for the challenging cancer treatment landscape.

Regulatory mechanisms in the immune system, known as immune checkpoints, help maintain a balance between activating and inhibiting immune responses in the body. In addition to preventing excessive immune reactions that can damage healthy tissues, they also play a crucial role in enabling the immune system to recognize and attack foreign invaders like pathogens and cancer cells [33]. Despite this, tumors have evolved mechanisms to manipulate immune checkpoints to prevent the activation of T cell effectors and to escape immune surveillance. Currently, cancer treatment extends far beyond traditional chemotherapy. Immunotherapy is on the rise as numerous studies show that it is effective across a wide range of solid tumor cancer types. In immunotherapy, immune checkpoint inhibitors are used to reactivate the immune system’s response against cancer cells. The available ICI drug classes include CTLA-4 inhibitors, PD-1/PD-L1 inhibitors, CTLA-4/PD-1 inhibitor combinations, and CTLA-4/LAG3 inhibitor combinations, with the response to these therapies being highly influenced by the type of cancer. A key determinant of ICI efficacy is the tumor microenvironment (TME), where immunotherapy is shown to be most beneficial in a highly infiltrated environment [33].

Several factors can contribute to the development of a suppressed TME that does not respond to immunotherapy, such as a low or absent expression of tumor antigens, inadequate antigen-presenting cells (APCs), as well as an immunosuppressive TME. In the TME, immune suppression is often mediated by suppressive immune cells, including tumor-associated macrophages (TAMs), cancer-associated fibroblasts (CAFs), regulatory T cells (Tregs), and myeloid-derived suppressor cells (MDSCs), all of which encourage angiogenesis and metastasis in the tumor. As a result of hypoxia, which is mediated by the HIF pathway, the immune suppressive environment is greatly influenced [32].

HIF-1α exerts its influence via hypoxia response elements (HREs) that bind to the promoters of various target genes, promoting their gene expression. HIF-1α also plays a direct role in upregulating immune checkpoint key players, including CTLA-4, LAG3, TIM3, and PD-L1 on tumor cells, as well as PD-1 and VISTA on MDSCs. The function of these checkpoints is to act as brakes on the immune system, dampening the activity of T-cells and aiding in the promotion of immunosuppression. Moreover, HIF-1α also has the capability of binding to the promoter of the PD-L1 gene, leading to the upregulation of this gene, further deactivating T cells and promoting the escape of the immune system in a manner that may lead to resistance against immune checkpoint inhibitors [32,34].

In addition to immune checkpoint inhibitors, HIF has also been found to affect other immunotherapies, such as VEGF inhibition via monoclonal antibody therapy. Because of hypoxia, one of the most important effects is the upregulation of VEGF, which stimulates angiogenesis, resulting in complicated, leaky, and poorly structured blood vessels, thus exacerbating hypoxia and resulting in increased cancer growth. In cancer treatment, angiogenesis can be inhibited by bevacizumab, a monoclonal antibody that targets VEGF, preventing the signaling pathway that results in the proliferation and migration of endothelial cells; however, HIF-1α may contribute to bevacizumab resistance. The HIF pathway can promote the development of blood vessels via alternative pathways such as platelet-derived growth factor (PDGF) and fibroblast growth factor (FGF). As a result, tumors can grow and metastasize despite bevacizumab’s inhibition of VEGF [32,35].

The role of HIF in anticancer drug resistance extends beyond immune checkpoints and the VEGF signaling pathway. HIF is also associated with multidrug resistance, particularly in a manner that promotes the upregulation of the ATP-binding cassette subfamily B member 1 (ABCB1). As an efflux drug transport protein, ABCB1 is located on the cell membrane and excretes drugs from tumor cells, thereby reducing drug accumulation and inhibiting apoptosis that may occur from chemotherapy. Both HIF-1α and ABCB1 are upregulated under hypoxic conditions, exacerbating drug resistance [32].

Furthermore, HIF-1α collaborates with other factors like TGF-β2 to activate genes such as GLI2, which enhance intrinsic tumor resistance to chemotherapeutic drugs. In various cancer types, HIF-1α has been implicated in drug resistance mechanisms, promoting tumor growth, invasion, and resistance to chemotherapy, including HIF-1α enhancement in gastric cancer, cisplatin resistance via autophagy enhancement in non-small cell lung cancer (NSCLC), oxaliplatin resistance in colon cancer, and sorafenib in hepatocellular carcinoma (HCC) via the upregulation of HIF-2α [32].

## 5. Role of HIFs in Solid Tumors

The following section describes the role of HIFs in various cancers.

### 5.1. Role of HIFs in Liver Cancer

Hepatocellular carcinoma (HCC), the predominant form of liver cancer, poses a global concern due to significant morbidity and mortality. Its asymptomatic early stages often lead to detection only during advanced phases, limiting treatment options. Intratumoral hypoxia, a hallmark of solid tumors including HCC, triggers metabolic adaptations mediated by HIFs, contributing to therapy resistance. Although sorafenib and immune checkpoint inhibitors (ICIs) are first-line treatments, drug resistance and the suppressive tumor microenvironment (TME) hinder their efficacy [36].

The TME displays immunosuppressive traits, leading to the exhaustion of anti-tumoral immune cells such as CD8+ cytotoxic T cells and natural killer (NK) cells. Elevated pro-tumoral immune cell presence includes M2-like macrophages, regulatory T cells (Treg), and myeloid-derived suppressor cells (MDSC). The immune checkpoint PD-1 restricts immune activity upon binding PD-L1, which cancer cells exploit to evade detection. Nivolumab, an anti-PD-1 monoclonal antibody disrupting PD-1/PD-L1 interaction, benefits only a minority (20%) of HCC patients [37]. Drug resistance emergence undermines existing treatment effectiveness.

Hypoxia-induced metabolic shifts drive resistance to HCC therapies. HCC cells’ adaptation to hypoxia involves HIF-regulated pathways, causing drug resistance. Sorafenib and some ICIs show partial efficacy, but complex TME limits patient benefits. Addressing hypoxic HCC drug resistance necessitates novel strategies targeting hypoxia-induced metabolic genes.

To combat resistance, the study suggests exploring combined therapies targeting HIF-induced metabolic pathways with TKIs and ICIs. Applicability beyond sorafenib resistance requires clarification [38]. Metabolic inhibitor interaction with immune function needs careful evaluation for safety and efficacy. The approach involves determining combined inhibitor effectiveness with TKIs or ICIs, aiming to overcome drug resistance and enhance HCC patient outcomes.

### 5.2. Role of HIFs in Renal Cell Carcinoma (RCC)

Renal cell carcinoma (RCC) is accountable for over 14,000 annual fatalities within the United States, primarily attributed to the progression of metastatic disease [39]. Prior to 2005, limited effective systemic interventions were available for managing patients with advanced RCC. RCC exhibits diverse histologic subtypes, with the most prevalent variant (>80%) identified as clear-cell RCC (ccRCC). ccRCC tumors consistently exhibit inactivating mutations or, less commonly, hypermethylation in both maternal and paternal copies of the VHL (von Hippel–Lindau) gene [40].

The VHL gene product, pVHL, constitutes an E3 ubiquitin ligase pivotal for oxygen sensing by facilitating degradation of the α-subunit of HIF. However, the loss or suppression of VHL leads to HIFα accumulation, consequently inducing transcription of hypoxia-responsive genes implicated in tumorigenesis, including VEGFA responsible for encoding VEGF (vascular endothelial growth factor). HIF-2α, a principal driver of ccRCC, controls several critical oncogenic pathways, positioning it as an optimal target for ccRCC treatment [41]. In hypoxic conditions, HIF-2α prevents proteasomal degradation by evading hydroxylation, and in pseudohypoxic states resulting from a loss-of-function mutation in pVHL, this evasion leads to the accumulation of HIF-2α. Subsequently, HIF-2α translocates to the nucleus and forms a heterodimer with HIF-1β, creating an active HIF transcription complex. This complex initiates the transcription of genes involved in tumorigenesis. Belzutifan, a small-molecule inhibitor, induces a conformational change in HIF-2α and impedes its heterodimerization with HIF-1β. Consequently, the assembly of the active HIF transcription complex is hindered [42].

Researchers have employed a genome-wide in vitro expression strategy to investigate the pathways affected by VHL loss in ccRCC. This approach led to the identification of Zinc fingers and homeoboxes 2 (ZHX2) as a target of VHL, where hydroxylation by VHL regulates the stability of the ZHX2 protein. Tumor cells from ccRCC patients with VHL loss-of-function mutations demonstrate increased abundance and nuclear localization of ZHX2. The depletion of ZHX2 hinders the growth of VHL-deficient ccRCC cells both in vitro and in vivo. Mechanistically, ZHX2 is implicated in promoting nuclear factor κB activation. These findings highlight ZHX2 as a potential therapeutic target for ccRCC [43].

### 5.3. Role of HIFs in Gastric Cancer

In the realm of gastric cancer, HIF’s significance emerges as it exerts a profound influence on the progression and development of this malignancy.

Central to HIF’s role in gastric cancer is its impact on cell behaviors that drive the disease’s pathogenesis. Firstly, HIF fuels the rapid proliferation of gastric cancer cells, essentially stoking the flames of their uncontrolled division. Via the activation of genes responsible for cell cycle progression, HIF contributes to the relentless growth of tumor cells within the stomach lining. Moreover, HIF’s involvement extends to facilitating the metastatic spread of gastric cancer cells. By triggering the epithelial–mesenchymal transition (EMT), a process enhancing cell mobility and invasiveness, HIF aids the migration of these cancer cells to distant sites within the body [44].

An intriguing aspect of HIF’s role in gastric cancer is its ability to confer resistance to cell death. This phenomenon, known as apoptosis resistance, enables cancer cells to withstand programmed cell death, enabling their survival and continued proliferation even under challenging conditions. Additionally, HIF plays a role in thwarting the efficacy of therapeutic interventions. By promoting the expression of genes that help cancer cells evade the effects of chemotherapy and targeted treatments, HIF contributes to drug resistance, a formidable obstacle in effective cancer management [45].

Small-molecule drugs mainly refer to organic compounds with molecular weights less than 1000, and they have been widely used and mature in theory. A few examples include apigenin, tipifarnib, dextran sulfate, and schisandrin. Many small-molecule drugs that inhibit the progression of gastric cancer via HIF have been found in basic experiments, while these drugs have not yet been clinically applied [46].

Given the success of Welireg, a drug targeting HIF-2 in treating VHL syndrome-related tumors, HIF plays a huge role in gastric cancer. We look forward to the future when scientists discover that HIF-targeting drugs for the treatment of gastric cancer will benefit patients in the clinic [47].

### 5.4. Role of HIFs in Breast Cancer

In breast cancer, the overexpression of HIF-1α is often associated with aggressive tumor characteristics. Studies have linked elevated HIF-1α expression to larger tumor sizes, higher tumor grades, increased proliferation markers (like Ki-67), and unfavorable molecular subtypes, such as HER2-positive and triple-negative breast cancers [48].

The role of HIF-1α in angiogenesis is particularly significant. HIF-1α induces the expression of vascular endothelial growth factor (VEGF), a key driver of angiogenesis. Furthermore, HIF-1α’s impact on therapy resistance adds another layer of complexity. The hypoxic microenvironment created by HIF-1α can lead to resistance against radiation therapy and some chemotherapy agents. The decreased oxygen levels in hypoxic regions of tumors can hinder the effectiveness of treatments, making those areas less susceptible to treatment-induced cell death [49].

The prognosis implications of HIF-1α in breast cancer are notable. Research has shown that higher levels of HIF-1α expression are associated with poorer survival outcomes, including decreased Disease-Free Survival (DFS) and Overall Survival (OS). These findings highlight the potential of HIF-1α as a prognostic biomarker, aiding in risk stratification and treatment planning for breast cancer patients [50].

Zinc Fingers and Homeoboxes 2 (ZHX2) are highly expressed in triple-negative breast cancer (TNBC). When ZHX2 is depleted, it hinders the growth and invasion of TNBC cells in vitro as well as orthotopic tumor growth and spontaneous lung metastasis in vivo. ZHX2 works by interacting with HIF-1α, boosting its activity in TNBC. Using advanced techniques, researchers found that ZHX2 collaborates with HIF-1α on specific gene promoters, promoting gene expression linked to cancer. Genes like AP2B1, COX20, KDM3A, or PTGES3L, coregulated by ZHX2 and HIF-1α, can partially reverse TNBC cell growth defects caused by ZHX2 depletion. Specific parts of ZHX2, denoted as residues (R491, R581, and R674), play a key role in controlling its function in TNBC cells. These findings highlight that ZHX2 activates cancer-promoting HIF-1α signaling, suggesting a potential therapeutic target for TNBC therapy [51].

## 6. HIF-Associated Future Cancer Therapies

As described above, HIF plays a pivotal role in regulating various genes that control processes such as angiogenesis, cell proliferation, and glucose metabolism. The dysregulation of HIF signaling and pathways has been associated with the development and progression of various cancers. Therefore, targeting HIF and its associated pathways is currently being explored for future anticancer therapy. There are several HIF-related mechanisms being studied for future cancer therapies, including HIF inhibitors, anti-angiogenic therapies, combination therapies, hypoxia-activated prodrugs, immunotherapy, biomarkers associated with HIF, gene editing, and RNA interference [52,53].

### 6.1. Reduction in HIF-1α Protein Synthesis

EZN-2968 is an anticancer agent that modulates the HIF-1α pathway. It works by inhibiting HIF-1α protein synthesis by blocking the translation of HIF-1α mRNA [54]. The goal of this trial was to evaluate the modulation of HIF-1α mRNA and tumor response of patients with refractory solid tumors. Ten patients were enrolled; two showed reductions in HIF-1α protein and mRNA levels, and one patient with a duodenal neuroendocrine tumor had prolonged stabilization of disease [54]. Although the trial was closed prematurely, it provided preliminary proof of HIF-1α mRNA modulation and potential antitumor activity.

Minnelide, a water-soluble pro-drug of triptolide, has been shown to be effective against pancreatic cancer in preclinical evaluation [55]. It works by downregulating the expression of p300, which prevents the assembly of the transcription complex of HIF-1α and decreases the transcriptional activity of HIF-1α [56]. It is currently in Phase II trials for the treatment of metastatic adenocarcinoma of the pancreas and is also being looked into for the treatments of gastrointestinal tumors, breast cancer, hepatocellular carcinoma, non-small cell lung carcinoma, acute myeloid leukemia, malignant mesothelioma, and gastric cancer [57].

Echinomycin is a small-molecule inhibitor of HIF-1α, which inhibits the activity of HIF-1 by binding to the HRE and blocking the binding of HIF-1 and HRE [58]. Although the clinical trials were stopped early due to being ineffective and toxic, there was the development of liposomal echinomycin, which could be less toxic and be used to inhibit tumor growth and metastatic disease [58].

EZN-2208, an active metabolite of irinotecan, suppresses HIF-1α at the mRNA level, therefore inhibiting the HIF-1α pathway. In a Phase 1 Trial, it was associated with clinical benefit in patients with neuroblastoma [59].

### 6.2. HIF-1 Inhibitors Destabilizing HIF-1α

Human histone deacetylases (HDACs) are a class of chromatin-modifying enzymes that are able to modify histone and non-histone proteins, leading to HIF-1α destabilization [60]. This class includes FK228 (Romidepsin), LBH589 (Panobinostat), SAHA (Vorinostat), and PXD-101 (Belinostat) [61]. Romidepsin suppresses HIF-1α stabilization and inhibits hypoxia-responsive angiogenesis factors [29]. Panobinostat inhibits HDAC, which causes cell cycle arrest and apoptosis, and clinical trials have shown the use of Panobinostat in combination with bortezomib and dexamethasone for the treatment of relapsed/refractory multiple myeloma [62]. Vorinostat is the first FDA-approved histone deacetylase inhibitor for the treatment of cutaneous T-cell lymphoma. It works by inhibiting HIF-1α via the acetylation of its associated chaperone, heat shock protein 90, and subsequently, downregulation of molecules, including GLUT 1 and VEGF [63]. Vorinostat is also currently being used in a clinical trial for the treatment of metastatic melanoma of the eye. These findings are summarized in Table 1.

### 6.3. HIF-1 Dimerization Inhibitor

Acriflavine binds to HIF-1α and HIF-2α directly, thus preventing HIF-1 dimerization and transcriptional activity. Acriflavine enhances the antitumor activity of sunitinib in a breast cancer model and of 5-fluorouracil used in the treatment of colorectal cancer much better than irinotecan [88].

## 7. Conclusions

HIF is a protein complex that plays a crucial role in the cellular response to low oxygen levels, known as hypoxia. HIF is involved in regulating various genes that control processes such as angiogenesis (the formation of new blood vessels), cell proliferation, and glucose metabolism. The dysregulation of HIF signaling has been implicated in the development and progression of various cancers. Understanding these mechanisms of HIF involvement in cancer is crucial for the development of targeted therapies that aim to disrupt HIF signaling or downstream pathways. There have been many past and current clinical trials looking at anticancer therapies involving the HIF pathways. It is important to note that while targeting HIF holds promise for future cancer therapies, there are significant challenges to overcome, such as the need for specificity to avoid interfering with normal tissue function. Ongoing clinical trials are evaluating the toxicities and efficacy of various HIF-targeted approaches and further research is necessary to completely understand the complex role of HIFs in cancer [7].

## Figures and Tables

**Figure 1 ijms-25-02060-f001:**
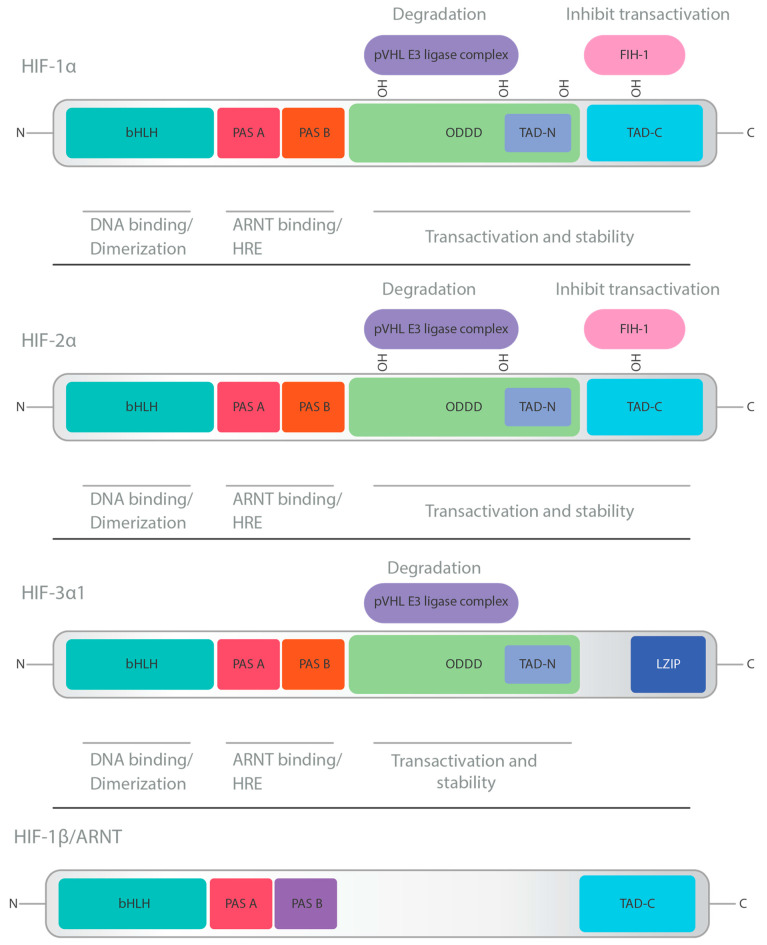
Schematic overview of the domain structures of hypoxia-inducible factor (HIF) members. The structural domains of HIF-1α, -2α, -3α and their transcriptional binding partner, HIF-1β/aryl hydrocarbon nuclear translocator (ARNT), together form the (HIF-α/HIF-1β) transcriptional complexes. The NH2-terminal of HIF-α and HIF-1b consists of basic helix–loop–helix (bHLH) and Per-ARNT-Sim homology (PAS) domains that are required for heterodimerization and DNA binding in the hypoxia response elements (HRE) at the target gene loci. The COOH-terminal of HIF-α contains two transactivation domains (TADs). The short half-life of HIF-α under nonhypoxic conditions is due to fast ubiquitination and proteasomal degradation. HIF-1α, -2α, and -3α also contain an oxygen-dependent degradation (ODD) domain. Factor-inhibiting HIF (FIH) also regulates HIF-α via oxygen-dependent hydroxylation. Only HIF-3α contains a leucine zipper (LZIP) domain in the COOH-terminal region. Adapted from [8].

**Figure 2 ijms-25-02060-f002:**
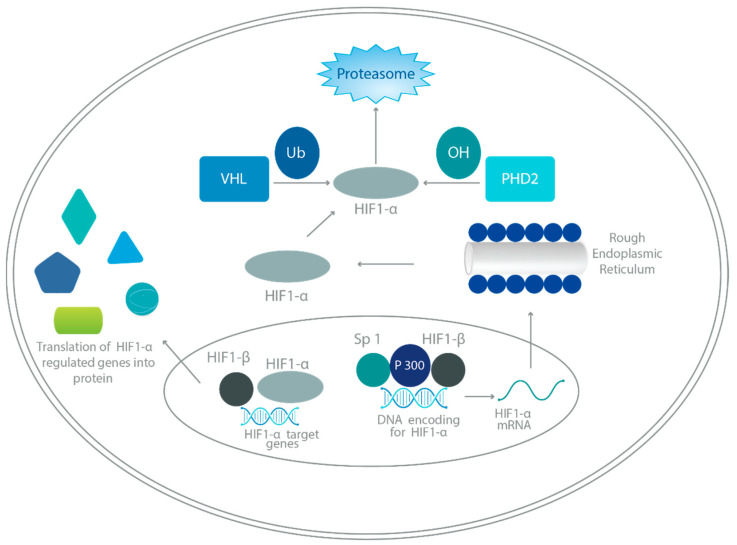
Hypoxia-inducible factor (HIF-1) pathway. The HIF-1α gene is transcribed in the nucleus with the help of specificity protein (Sp) 1, P300, and HIF-1β. Once translated in the cytoplasm, the HIF-1α protein can be hydroxylated and ubiquitinated, in which case it will be degraded by proteasomes (under normal oxygen conditions). In the setting of hypoxia, it can re-enter the nucleus and form a transcription complex with the HIF-1β subunit. If successfully stabilized with the latter subunit, the final complex ultimately will function to regulate target genes such as vascular endothelial growth factor and cathepsin D. Abbreviations: PHD: proline-hydroxylase domain-containing molecules; Ub: ubiquitin; VHL: von Hippel–Lindau protein. Adapted from [7,14].

**Table 1 ijms-25-02060-t001:** Current clinical status evaluating HIF-1α inhibitors as potential anticancer therapies, adapted from [29].

Mechanism of HIF-1 Inhibition	HIF-1 Inhibitor	Cancer Type	Reference
Decreasing HIF-1α mRNA expression	EZN-2208 (PEG-SN38)	Refractory solid tumors Metastatic colorectal cancer	[59,64,65]
Decreasing HIF-1α protein synthesis	EZN-2968	Refractory solid tumors	[54]
Decreasing HIF-1α stabilization	FK228 (Romidepsin)	Non-small cell lung carcinoma (NSCLC) Head and neck cancer	[66,67]
	LBH589 (Panobinostat)	Multiple myeloma (MM) Refractory solid tumors Refractory MM	[68,69,70]
	Vorinostat	Glioblastoma Advanced melanoma Mesothelioma	[71,72,73]
	17-allylamino-17-demethoxygeldanamycin (17-AAG) (Tanespimycin)	Prostate cancer Refractory MM	[74,75]
	SCH66336 (Lonafarnib)	Squamous cell carcinoma of the head and neck (SCCHN) Lung carcinoma Colorectal cancer	[76,77,78]
Decreasing HIF-1/ DNA binding	Echinomycin	Ovarian cancer Breast cancer Renal cell carcinoma Colorectal cancer	[79,80,81,82]
Decreasing HIF-1α protein synthesis and transcriptional activity	2-methoxyestradiol (2ME2/Panzem)	MM Prostate cancer Ovarian cancer Carcinoid tumors	[83,84,85,86]
Decreasing HIF-1α and HIF-2α protein synthesis and transcriptional activity	32-134D	Hepatocellular carcinoma	[87]

## Data Availability

Not applicable.

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
