# Peer review of "Therapeutic Targeting of Hypoxia-Inducible Factors in Cancer"

_ijms, 2024, doi:10.3390/ijms25042060_

Round 1

Reviewer 1 Report

Comments and Suggestions for Authors

Hypoxia-inducible factors (HIFs) activate the transcription of genes that are involved in crucial aspects of cancer biology and inhibition of HIFs activity could be very efficient anti-cancer therapeutics. In this manuscript, the authors summarized the role and mechanisms of HIF in cancer progression and HIF associated cancer therapies. Overall, this is an interesting topic. My specific comments are listed below.

1.      Both figures ware adapted from Ziello’s work (Yale J Biol Med, 2007) which was published 16 years ago. I strongly suggest the authors should make new diagrams that include recent progress about HIF research. And in figure 2, “Translation of HIF1-α regulated enes into proteins” should be “Translation of HIF1-α regulated genes into proteins”.

2.      Considering there are lots of review papers that discussed HIF functions in cancers and HIF associated cancer therapies, to include the most recent progress is required for a meaningful review study. So, some recent studies should be discussed in this manuscript, such as “Science. 2018 Jul 20;361(6399):290-295. eLife. 2021; 10: e70412; Nat Commun. 2022 Jan 14;13(1):316”.

3.      As a review paper, it is highly recommended that the authors should combine their own related research work with other literatures to give some perspective comments.

4.      A HIF-2α inhibitor, Belzutifan (PT2977), was approved by FDA to treat patients with advanced kidney cancer. This is an important event in the field of therapeutic targeting of HIF-1 in cancers and should be mentioned in this study.

5.       In Table 1, the references should be listed in the table along with the corresponding HIF-1α inhibitors. And a new HIF-1 inhibitor, 32-134D ((J Clin Invest. 2022 May 2;132(9):e156774) should be included.

Comments on the Quality of English Language

The quality of English language is good.

Author Response

Comments 1: Both figures ware adapted from Ziello’s work (Yale J Biol Med, 2007) which was published 16 years ago. I strongly suggest the authors should make new diagrams that include recent progress about HIF research. And in figure 2, “Translation of HIF1-α regulated enes into proteins” should be “Translation of HIF1-α regulated genes into proteins”.

Response 1: Thank you for pointing this out. We agree with this comment. Therefore, we have updated the figure 1 and included diagrams for HIF structures. The citation has been updated to a latest citation. For figure 2, we have corrected the spelling mistake and updated citations.

Comments 2: Considering there are lots of review papers that discussed HIF functions in cancers and HIF associated cancer therapies, to include the most recent progress is required for a meaningful review study. So, some recent studies should be discussed in this manuscript, such as “Science. 2018 Jul 20;361(6399):290-295.eLife. 2021; 10: e70412; Nat Commun. 2022 Jan 14;13(1):316”.

Response 2: Agree. We have, accordingly added recent studies. Refer to Section 5.2 Role of HIFs in Renal cell carcinoma (RCC) (Last Paragraph Line: 322-329) and Refer to Section 5.4 Role of HIFs in Breast Cancer (Last Paragraph Line: 376-386)

Comments 3: As a review paper, it is highly recommended that the authors should combine their own related research work with other literatures to give some perspective comments.

Response 3: The team has added relevant research and does not have any additional related research to be added.

Comments 4: A HIF-2α inhibitor, Belzutifan (PT2977), was approved by FDA to treat patients with advanced kidney cancer. This is an important tevent in the field of therapeutic targeting of HIF-1 in cancers and should be mentioned in this study.

Response 4: Agree. We have added recent studies. Refer to Section 5.2 Role of HIFs in Renal cell carcinoma (RCC) (Line: 318-321)

Comments 5: In Table 1, the references should be listed in the table along with the corresponding HIF-1α inhibitors. And a new HIF-1 inhibitor, 32-134D ((J Clin Invest. 2022 May 2;132(9):e156774) should be included.

Response 5: Agree. We have added to Table 1.

Reviewer 2 Report

Comments and Suggestions for Authors

The review concerns the therapeutic targeting of HIF1 in cancer. In particular, the authors declare that the main role of HIF1 in the regulation of angiogenesis, cell proliferation, and glucose metabolism will be explored in depth and that emphasis will be given to the potential of HIF-targeted therapies in the disruption of cancer-related signal pathways. However, the review appears rather superficial and would need to be substantially improved to be published.

Some suggestions:

1.       The same concepts are often repeated throughout the text (for example the definition of hypoxia or angiogenesis, the fact that the HIF1 alpha subunit is oxygen-sensitive, and so on);

2.       In “HIF structure and physiology” only HIF1 alpha is described, but no informations about HIF2 and HIF3 alpha are included

3.       Other oxygen-independent mechanisms of HIF1alpha expression should be cited

4.       In my opinion, the role of HIF in the different types of cancer should be treated separately without subparagraph

5.       A column should be added to Table 1, containing the relevant bibliography. Furthermore, the name of all the inhibitors should be reported

6.       In the text, all spaces between the sentences must be eliminated

Author Response

Comments 1: The same concepts are often repeated throughout the text(for example the definition of hypoxia or angiogenesis, the fact that the HIF1 alpha subunit is oxygen-sensitive, and so on)

Response 1: Thank you for pointing this out. We agree with this comment. Repetition has been minimized throughout this revised article

Comments 2: In “HIF structure and physiology” only HIF1 alpha is described, but no informations about HIF2 and HIF3 alpha are included

Response 2: Agree, we have added more information on HIF 2 and HIF 3 in Section 2. HIF Structure and Physiology (line 96-107)

Comments 3: Other oxygen-independent mechanisms of HIF1alpha expression should be cited

Response 3: Section 4 has been reviewed with reference to oxygen-independent mechanisms of HIF1 alpha removed.

Comments 4:  In my opinion, the role of HIF in the different types of cancer should be treated separately without subparagraph

Response 4: Thanks for this feedback, a new section 5 has been created to explain different types of cancer.

Comments 5:  A column should be added to Table 1, containing the relevant bibliography. Furthermore, the name of all the inhibitors should be reported

Response 5: A column is added for bibliography and names have been added where found.

Comments 6: In the text, all spaces between the sentences must be eliminated

Response 6: Thank you. All spaces are removed where found.

Round 2

Reviewer 1 Report

Comments and Suggestions for Authors

The authors revised their manuscript and provided relatively reasonable explanations for my questions. Some details were also added into this manuscript. The data in this study are integrated and more convincing. This manuscript should be able to bring the readers some understanding about the role of Hypoxia-Inducible Factor (HIF-1) in cancers. I have no more questions for the current version.

Author Response

Thank you for your approval of the revised manuscript

Reviewer 2 Report

Comments and Suggestions for Authors

The manuscript could be accepted in the current form

Author Response

(The authors gave the same response as above.)
